# Human milk cortisol and immune factors over the first three postnatal months: Relations to maternal psychosocial distress

Marina Aparicio[1☉], Pamela D. Browne[2☉], Christine Hechler[3], Roseriet Beijers[2,3], Juan Miguel Rodríguez[1], Carolina de Weerth[2‡]*, Leonides Fernández[4‡]*

1 Department of Nutrition and Food Science, Faculty of Veterinary Sciences, Complutense University of Madrid, Madrid, Spain, 2 Department of Cognitive Neuroscience, Donders Institute for Brain, Cognition and Behaviour, Radboud University Medical Center, Nijmegen, The Netherlands, 3 Developmental Psychology, Behavioural Science Institute, Radboud University, Nijmegen, The Netherlands, 4 Departmental Section of Galenic Pharmacy and Food Technology, Faculty of Veterinary Sciences, Complutense University of Madrid, Madrid, Spain

☉ These authors contributed equally to this work.
‡ These authors also contributed equally to this work.
* Carolina.deWeerth@radboudumc.nl (CdW); leonides@ucm.es (LF)

**Data Availability Statement:** All relevant data are within the manuscript and its Supporting Information files.

## Abstract

### Background

Many biologically active factors are present in human milk including proteins, lipids, immune factors, and hormones. The milk composition varies over time and shows large inter-individual variability. This study examined variations of human milk immune factors and cortisol concentrations in the first three months post-partum, and their potential associations with maternal psychosocial distress.

### Methods

Seventy-seven healthy mothers with full term pregnancies were enrolled, of which 51 mothers collected morning milk samples at 2, 6 and 12 weeks post-delivery. Maternal psychosocial distress was assessed at 6 weeks post-delivery using questionnaires for stress, anxiety, and depressive symptoms. Immune factors were determined using multiplex immunoassays and included innate immunity factors (IL1β, IL6, IL12, IFNγ, TNFα), acquired immunity factors (IL2, IL4, IL10, IL13, IL17), chemokines (IL8, Groα, MCP1, MIP1β), growth factors (IL5, IL7, GCSF, GMCSF, TGFβ2) and immunoglobulins (IgA, total IgG, IgM). Cortisol was quantified using liquid chromatography-tandem mass spectrometry. A linear mixed effects model was fit to test whether stress, anxiety, and depressive symptoms individually predicted human milk cortisol concentrations after accounting for covariates. Repeated measurement analyses were used to compare women with high (n = 13) versus low psychosocial distress (n = 13) for immune factors and cortisol concentrations.

### Results

Virtually all immune factors and cortisol, with the exception of the granulocyte-macrophage colony-stimulating factor (GMCSF), were detected in the human milk samples. The

**Funding:** LF and JR's work was funded by project AGL2016-75476-R from the Ministerio de Economía y Competitividad (Spain). RB's work was funded by a VENI grant from the Netherlands Organization for Scientific Research (016. Veni.195.197). A Jacobs Foundation Advanced Research Fellowship and a VICI grant from the Netherlands Organization for Scientific Research (016.Vici. 185.038) supported CdW's work. The funders had no role in study design, data collection and analysis, decision to publish, or preparation of the manuscript.

**Competing interests:** The authors have declared that no competing interests exist.

concentrations of the immune factors decreased during the first 3 months, while cortisol concentrations increased over time. No correlation was observed between any of the immune factors and cortisol. No consistent relationship between postnatal psychosocial distress and concentrations of immune factors was found, whereas higher psychosocial distress was predictive of higher cortisol concentrations in human milk.

## Conclusion

In the current study we found no evidence for an association between natural variations in maternal distress and immune factor concentrations in milk. It is uncertain if this lack of association would also be observed in studies with larger populations, with less uniform demographic characteristics, or with women with higher (clinical) levels of anxiety, stress and/or depressive symptoms. In contrast, maternal psychosocial distress was positively related to higher milk cortisol concentrations at week 2 post-delivery. Further investigation on maternal psychosocial distress in relation to human milk composition is warranted.

## Introduction

Many bioactive factors are present in human milk, including immune factors [1] and hormones [2,3]. These factors contribute to optimal infant health and development [4]. The immune factors in human milk complement the infant's immature immune system [5,6]. In addition to anti-infectious properties, immune factors also demonstrate anti-inflammatory properties and play a role in the establishment of the infant's gut barrier and gut microbiota. The latter favors the development of the infant's intestinal and immune functioning [7,8,9]. Concentrations of immune factors tend to be higher in colostrum compared to mature milk, with the decrease occurring during the first months postpartum [10,11]. However, most of the available studies only assessed a narrow panel of immune factors [1,11, 12].

The immunological composition of human milk varies greatly within an individual mother over time, but also between women [1]. This variation seems partly explained by different maternal factors, including maternal postnatal psychosocial distress (henceforth referred to as psychosocial distress) [10,13,14,15,16]. In the present study, psychosocial distress is defined as higher levels of stress, anxiety and depressive symptoms during the postpartum period. It differs from postpartum blues in that it can last for over 3 months instead of the first week after delivery [17]. Moreover, unlike postpartum depression, psychosocial distress is not necessarily diagnosed by clinical evaluation [18]. Psychosocial distress is highly prevalent, with up to 25% of women experiencing symptoms of distress after delivery [19]. Hypothetically, a state of psychological distress may modulate the maternal immune system, including the mucosa-associated lymphoid tissue (MALT) and plasma cells in the mammary gland. Indeed, maternal postpartum depression has been associated with depressed cellular immunity [13]. Modulations in the maternal immune system may consequently lead to shifts of immune factor concentrations in human milk [5]. In line with this, a previous study with 50 women found that maternal perceived stress was correlated with human milk secretory immunoglobulin A (sIgA) concentrations [5], and higher levels of depressive symptoms in 139 mothers have been associated with higher concentrations of transforming growth factor-beta (TGFβ) in human milk [16].

Recently, in the same sample of women as included in the current study, we found that human milk cortisol concentrations increased from week 2 to week 12 [20]. In the present study, we determined whether maternal distress was related to higher cortisol concentrations in human milk. Cortisol is the hormonal end product of the hypothalamic-pituitary-adrenal axis (HPA-axis), the stress control system. Exposure to higher levels of human milk cortisol may influence infant behavior and brain development [3,21]. Animal studies showed that serum cortisol concentrations increased during physical and psychological distress, leading to increased concentrations of milk cortisol [22,23]. In humans, relaxation therapy was effective in lowering milk cortisol at two weeks postpartum [24]. Other observational studies that examined whether cortisol concentrations (i.e. milk, serum or salivary cortisol) were related to maternal distress have shown conflicting results [3,14,24,25,26,27].

The present study sought to shed light on the possible relations between maternal psychosocial distress, immune factors, and cortisol in human milk in the early postpartum period. The first aim was to longitudinally investigate the presence, concentrations, and potential changes over time of a relatively large panel of immune factors in human milk during the first three months postpartum. The immune factors included innate immunity factors (IL1β, IL6, IL12, IFNγ, TNFα), acquired immunity factors (IL2, IL4, IL10, IL13, IL17), chemokines (IL8, Groα, MCP1, MIP1β), growth factors (IL5, IL7, GCSF, GMCSF, TGFβ2), and immunoglobulins (IgA, total IgG, IgM). The second aim was to identify whether maternal postnatal psychosocial distress (i.e. perceived stress due to daily hassles, general anxiety, and depressive symptoms) in a group of 51 healthy mothers was related to changes in the presence and concentration of immune factors and cortisol in human milk.

## Materials and methods

### Participants and procedures

This project is part of the BINGO study (Dutch acronym for *Biological Influences on Baby's Health and Development*), which is a longitudinal study designed to identify prenatal and early postnatal predictors of infant health and development. The Ethics Committee of the Faculty of Social Sciences, Radboud University approved the study (ECSW2014-1003-189). All participants gave written informed consent prior to the start of the study.

Participants were recruited for the BINGO study via the project's website and folders that were handed out at various locations in the region Nijmegen-Arnhem, including midwife practices, pregnancy courses, and baby stores. Initial exclusion criteria were an unhealthy, complicated pregnancy, insufficient mastery of the Dutch language, excessive alcohol use (i.e., alcohol dependency) and/or drug use. Women who were interested in potential participation received additional study information by mail. Ninety-eight expectant women signed up for the study by providing written informed consent, 88 of these women were eligible for study participation, and participated in this project (S1 Fig). The exclusion criteria after birth were: gestational age at birth <37 weeks, 5-minute Apgar score <7, birth weight <2,500 g, congenital malformations, complications during pregnancy after initial contact, maternal antibiotic use, incomplete milk samples (i.e. not collected at all three time points), and maternal illness. Maternal illness was defined as any general illnesses (i.e. temporary illnesses such as fever during the preceding week, or chronic diseases) and pregnancy-related illnesses that required second line obstetric care. These criteria rendered a final group of 51 mothers for the analyses (S1 Fig).

Mothers filled out paper questionnaires on psychosocial stress, anxiety, and depressive symptoms at home at six weeks postpartum. Milk samples were collected at 2, 6 and 12 weeks post-delivery.

## Collection of human milk samples

Mothers collected milk after the infant reached the age of 2 weeks [mean (SD) age of 14.75 (1.84) days], 6 weeks [mean (SD) age of 43.58 (5.02) days], and 12 weeks [mean (SD) age of 85.35 (2.33) days]. Prior to collection, mothers washed their hands, breasts, and nipples, and cleaned the small collection cups with boiling water. Subsequently, prior to feeding the infant, they collected 20 mL of the first foremilk in the morning [mean (SD) time at 08:36 (02:48) am] by hand expression. Additionally, mothers noted the date and time of sample collection as well as maternal illnesses and/or intake of medication the preceding week. After collection, samples were immediately stored in the mothers' freezers (-18 - -20 ºC). When the infant was 13 weeks of age, samples were collected in a portable freezer to be stored at -80 ºC at Radboud University, the Netherlands. Aliquots of the samples were afterwards shipped by temperature-controlled shipment to the Complutense University of Madrid, Spain (for immune factor analyses), and to the Utrecht University Medical Centre, the Netherlands (for cortisol analyses).

## Immunological analysis of human milk

Immune factors in the milk samples were determined at the Complutense University of Madrid (Spain) following the procedure described by Ruiz et al. [1]. The concentrations of innate immune factors (IL1β, IL6, IL12, IFNγ, TNFα), acquired immunity factors (IL2, IL4, IL10, IL13, IL17), chemokines (IL8, Groα, MCP1, MIP1β), and growth factors (IL5, IL7, GCSF, GMCSF, TGFβ2) were determined by magnetic beads-based multiplex immunoassays, using a Bio-Plex 200 instrument (Bio-Rad, Hercules, CA) and the Bio-Plex Pro Human Cytokine, Chemokine, and Growth Factor Assays and the Bio-Plex Pro TGF-β 3-plex Immunoassay kits (Bio-Rad). Epidermal growth factor (EGF) was determined by ELISA using the RayBio Human EGF ELISA kit (RayBiotech, Norcross, GA). The concentrations of immunoglobulins (IgA, total IgG, and IgM) in the milk samples were determined using the Bio-Plex Pro Human Isotyping Assay kit (Bio-Rad) in the Bio-Plex system instrument.

Prior to their analysis, the samples (1 mL) were processed and aliquoted as described previously [28]. Every assay was run in duplicate according to the manufacturer's instructions and standard curves were performed for each analyte.

## Human milk cortisol concentration analysis

Cortisol concentrations in milk were determined at the Clinical Chemistry and Haematology laboratory (LKCH) of the University Medical Centre in Utrecht, the Netherlands. Cortisol was quantified by Liquid Chromatography-tandem Mass Spectrometry (LC-MS/MS) adding cortisol-$D_4$ as an internal standard, as described in Hechler et al. [20]. The limit of quantification for cortisol was 1.0 nmol/L.

## Stress

General stress was measured with the Alledaagse Problemen Lijst (Everyday Problem Checklist; EPL), which assessed the occurrence and intensity of daily hassles [29]. Participants indicated whether a daily hassle (49 items presented) had occurred in the past two months and how much it had bothered them on a 4-point Likert scale ranging from 1 ("I do not mind at all) to 4 ("I do mind a lot"). The mean intensity rating of daily hassles was computed by dividing the sum of total (negative) valence by the frequency of the events, with higher values indicating more experienced negativity. Internal consistency in this sample was good, with Cronbach's α equal to 0.84.

## Anxiety

A Dutch translation of the State Trait Anxiety Inventory (STAI-S) was used to measure maternal postnatal state anxiety [30]. The STAI-S consists of 20 statements related to feelings of anxiety at the present moment. Participants scored on a 4-point scale from 1 ("not at all") to 4 ("very much") how they felt at a specific moment and scores were summed up. Higher scores indicated more feelings of anxiety. Results indicated excellent internal consistency in this sample ($\alpha$ = 0.90).

## Depressive symptoms

Depressive symptoms were measured with the Edinburgh Postnatal Depression Scale (EPDS) [31]. The EPDS consists of 10 items for which participants indicated whether they experienced depressive symptoms in the past seven days. Responses were scored on a 4-point scale ranging from 0 to 3 in seriousness of symptoms; higher scores indicated more depressive symptoms. Internal consistency was good in this sample with $\alpha$ = 0.87.

## Psychosocial distress

Pregnancy psychosocial distress is a multidimensional concept of different pregnancy specific and non specific mood states [32,33]. Therefore, as we did in our previous studies, we aggregated the scores of the three distress questionnaires on stress, anxiety and depressive symptoms into one measure of "maternal psychosocial distress" [34,35,36].

## Statistical analyses

Normality of data distribution was examined through visual inspection of histograms and Shapiro-Wilks tests. Variables following a normal distribution were expressed as the mean and standard deviation or 95% confidence interval (CI) of the mean, while those variables that were not normally distributed were expressed as the median and interquartile range (IQR). Cortisol and immune factor concentrations were logarithmically transformed prior to analysis when indicated. Postnatal week and time of milk collection were included in the analyses as potential confounders.

The strength and direction of association between variables (cortisol and immune factor concentrations in milk and the scores of stress, anxiety, and depressive symptoms) was measured using the Spearman's rank-order correlation analyses. The correlation matrix was visualized using R package "*corrplot*" [37].

Principal Component Analyses (PCA) with a variable reduction approach ($\cos^2 > 0.2$) was performed to test for similarities among milk samples according to the concentrations of immune factors and cortisol as active variables using "FactoMineR" package in R [38]. Postnatal week and psychosocial distress level were used as supplementary variables to label milk samples in the scatterplot.

Kruskal-Wallis tests were used in non-parametric analysis, followed by *post-hoc* Nemenyi-test when required, and one-way ANOVA was applied for parametric data. Friedman's non-parametric repeated measures comparisons followed by *post-hoc* Nemenyi tests were applied to evaluate differences in the concentrations of immune factors and cortisol at the three postnatal sampling times. Comparisons of cortisol concentrations in milk, time of collection during the day and time-elapse between wake up and milk sampling were performed with *t*-tests. Differences in the detection frequencies of the immune factors and of the number of samples collected at each of the three sampling points were evaluated by the Chi-squared or Fisher's exact test, followed by a *post-hoc* Nemenyi-test adjusted to $\chi^2$ statistics for pairwise multiple

comparison, when required. The PMCMRplus package in R was used to perform these analyses [39].

Linear mixed effects model of log-transformed concentrations of cortisol and immune factors including postnatal week, sample collection hour, time-elapse between wake up and milk sampling, and the scores of stress, anxiety, and depressive symptoms as effects were run using StatGraphics Centurion 18 (version 18.1.06) (Statpoint Technologies, Inc., The Plains, VA, USA).

All statistical analyses were considered significant at the $p < 0.05$ level.

## Results

### Participant characteristics

Women participating were between 26 and 40 years of age, healthy, predominantly highly educated (80%), and with an uncomplicated, full term pregnancies (Table 1). The mean (SD) gestational age at delivery was 39.73 (1.66) weeks. Maternal scores on stress, anxiety and depressive symptoms are shown in Table 1.

### Relationship between cortisol and immune factors in human milk

Given that immune factors and cortisol concentrations were analyzed in parallel, potential correlations between these compounds were tested. Cortisol concentrations were not correlated with the concentrations of the most frequently detected immune factors assayed (Spearman's rank correlation: $-0.19 \leq r_s \leq 0.11$) (Fig 1). In contrast, most of the immune factors were positively correlated, and the correlation was particularly strong between the pairs IL8 and MIP1β, and IL7 and Groα at weeks 2 and 6 ($r_s > 0.75$) (Fig 1).

**Table 1. Maternal and infant characteristics of all participants (n = 51) included in the study.**

| **Maternal characteristics** | |
|---|---|
| Age (years) | |
| Mean (SD) | 32.04 (3.70) |
| Range (min, max) | 26.00–40.00 |
| Educational background | |
| Middle job training, n (%) | 8 (16) |
| Special college, n (%) | 15 (29) |
| University, n (%) | 26 (51) |
| Other, n (%) | 2 (4) |
| Psychosocial distress | |
| Postnatal stress[a], median (IQR) | 2.14 (1.83–2.50) |
| Postnatal general anxiety[b], median (IQR) | 27.0 (23.0–32.0) |
| Postnatal depressive symptoms[c], median (IQR) | 5.0 (3.0–7.0) |
| **Infant characteristics** | |
| Infant gender | |
| Boy, n (%) | 25 (49) |
| Girl, n (%) | 26 (51) |
| Gestational age at birth (weeks) | |
| Mean (SD) | 39.73 (1.66) |
| Range (min, max) | 35.57–42.14 |

[a] Alledaagse Problemen Lijst (Everyday Problem Checklist; EPL; scale: 0–6) [29].

[b] State Trait Anxiety Inventory (STAI-S; scale: 20–80) [30].

[c] Edinburgh Postnatal Depression Scale (EPDS; scale: 0–30) [31].

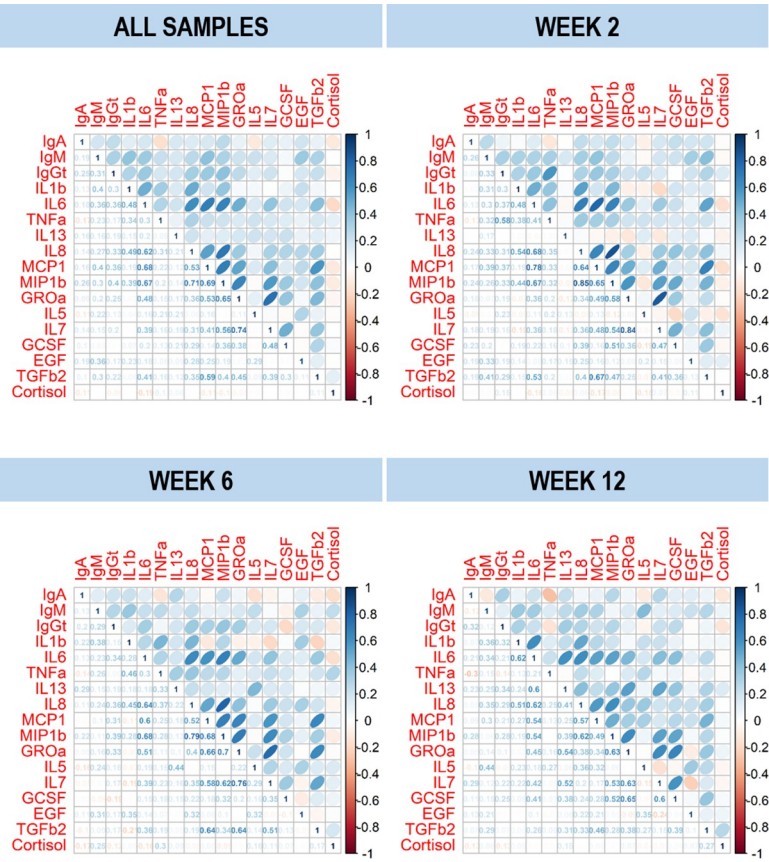

**Fig 1. Correlation of cortisol and immune factors concentrations in milk samples.** Spearman's rank correlation matrix of cortisol and immune factors concentrations in milk samples provided by women participating in this study (n = 51). Blue color indicates positive correlation and red color negative correlation. Color intensity is proportional to the magnitude of correlation coefficients. The actual values of the Spearman's rank correlation coefficient ($r_s$) are shown in the lower left corner of the plots.

In order to find potential pattern profiles for immune factors and cortisol concentrations in the samples, an initial PCA exploratory analysis was performed. PCA analysis did not reveal any distinct patterns in the concentrations of cortisol and immune factors in the samples. Samples roughly distributed according to the three postnatal weeks in the scatter plot (S2 Fig). The explained variance using the first two dimensions was about 33%, being IL8 the most influential variable (explaining 21.34% of the variance; S2 Fig). Samples did not cluster when labelled with the distress level (S2 Fig).

## Concentrations and changes over time of cortisol

Median (IQR) cortisol concentration in milk was 7.93 (4.22–14.25) μg/dL and ranged from 1.43 μg/dL to 64.20 μg/dL during the first three months postpartum (Fig 2). Although more than half of the samples (n = 85) were obtained between 06:00 and 09:00, there was a large range of collection times (from 03:00 to 23:00) (Fig 2). The association between the time of milk collection and the cortisol concentrations is shown in Fig 2, where samples for which sampling time was registered were grouped at hourly intervals. Most samples (n = 139) were taken between 05:00 and 12:00. The highest mean cortisol concentrations were recorded in samples taken between 06:00 and 06:59. Cortisol concentrations continuously increased in the

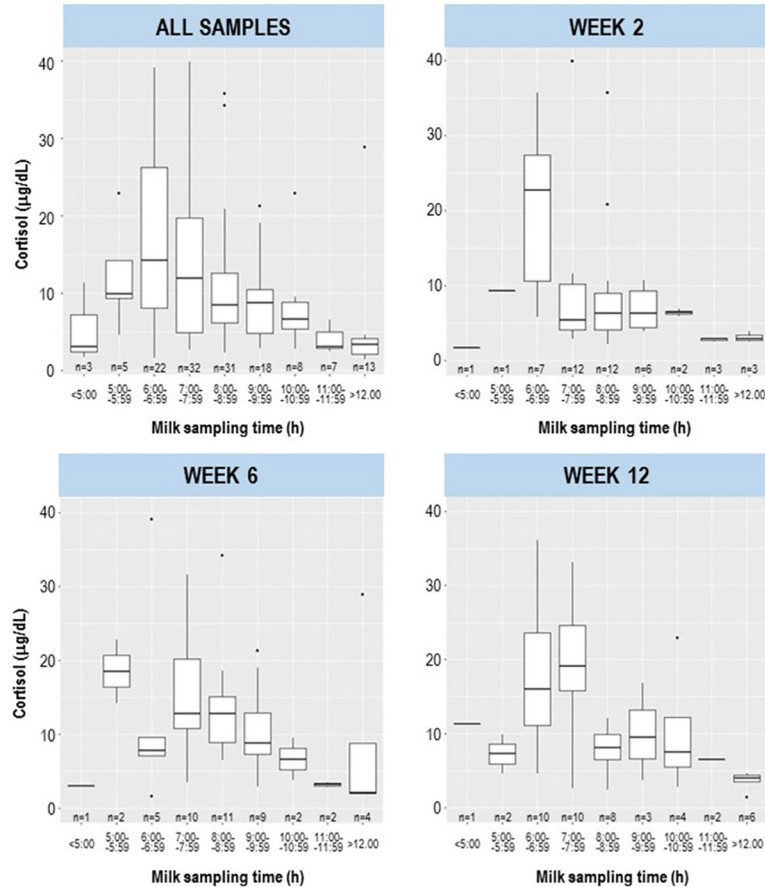

**Fig 2. Cortisol concentration in milk samples of participants during the day at all postnatal times (all samples), and 2, 6 and 12 weeks postpartum.** Box plots represent the cortisol concentration of samples grouped in hourly intervals from 5.00 am to 12.00 pm; all samples collected before 05.00 am and after 12.00 pm were grouped separately. The number of samples included in each boxplot is indicated under each boxplot. White circles represent outliers ($>1.5\times$ IQR). Some samples (n = 13) were not included in the graphs because the collection time was not available.

early morning from 05:00 to 06:00–06:59, and showed a continuous decrease throughout the day from 06:59 until noon.

Mean cortisol concentration in samples taken at week 2 (6.61 µg/dL) was lower than concentrations at weeks 6 and 12 (8.51 and 9.12 µg/dL, respectively) (S3 Fig). The difference in concentrations was statistically significant between weeks 2 and 12 (p = 0.002) (S3 Fig). The number of samples collected at each of the three sampling points (week 2, 6 and 12) were similarly distributed among the different hourly intervals (Fig 2; $\chi^2$ statistical test for a 3×9 contingency table; $\chi^2$ = 8.59, p = 0.929). This excludes that the observed increase of cortisol concentrations over time is due to differences in milk sampling times.

The diurnal rhythm of cortisol in milk was also observed when the samples were analyzed separately for the three postnatal weeks (Fig 2). At week 2, the median cortisol concentrations were highest (22.72 µg/dL) in samples collected at 06:00–06:59, although the variability was high (between 5.79 and 35.66 µg/dL). In contrast, at weeks 6 and 12, the highest median cortisol concentration was found in samples collected at 05:00–05:59 (18.55 µg/dL) and 07:00–07:59 (19.04 µg/dL), respectively (Fig 2).

**Table 2. Frequency and concentration of immune factors in milk samples.**

| Immune factor | Week 2 | | | Week 6 | | | Week 12 | | | |
|---|---|---|---|---|---|---|---|---|---|---|
| | n (%)[a] | Median (IQR)[b] | n (%) | Median (IQR) | n (%) | Median (IQR) | n (%) | Median (IQR) | p–value[c] | p–value[d] |
| *Immunoglobulins* | | | | | | | | | | |
| IgA (g/L) | 51 (100) | 1.68 (1.08–2.09)[a] | 51 (100) | 1.26 (0.92–1.67)[b] | 51 (100) | 1.26 (0.83–1.68)[b] | 1.000 | <0.001 |
| IgGt (mg/L) | 51 (100) | 52.10 (39.10–76.42)[a] | 51 (100) | 43.60 (32.64–57.71)[b] | 51 (100) | 42.00 (31.51–56.02)[b] | 1.000 | <0.001 |
| IgM (mg/L) | 51 (100) | 64.73 (47.84–97.12)[a] | 51 (100) | 38.19 (21.73–61.92)[b] | 51 (100) | 33.06 (16.93–49.22)[c] | 1.000 | <0.001 |
| *Innate immunity* | | | | | | | | | | |
| IL1β (ng/L) | 44 (86) | 0.52 (0.31–1.36)[a] | 35 (69) | 0.51 (0.29–0.87)[b] | 34 (67) | 0.37 (0.18–0.69)[b] | 0.046 | 0.065 |
| IL6 (ng/L) | 28 (55) | 9.36 (5.31–19.63)[a] | 16 (31) | 5.32 (3.09–12.36)[b] | 11 (22) | 2.64 (1.19–7.42)[b] | 0.002 | 0.002 |
| IL12(p70) (ng/L) | 2 (4) | 2.41 (1.44–3.39) | 0 | – | 2 (4) | 0.44 (0.42–0.45) | 0.547* | – |
| IFNγ (ng/L) | 4 (8) | 8.53 (6.81–12.64) | 2 (4) | 6.65 (5.01–8.29) | 2 (4) | 2.49 (2.31–2.66) | 0.600* | – |
| TNFα (ng/L) | 38 (75) | 2.69 (1.38–3.77) | 34 (67) | 2.24 (1.38–3.77) | 33 (65) | 2.84 (1.38–4.08) | 0.530 | 0.914 |
| *Acquired immunity* | | | | | | | | | | |
| IL2 (ng/L) | 1 (2) | 0.93 | 2 (4) | 0.92 (0.69–1.14) | 2 (4) | 3.41 (2.70–4.11) | 0.873* | – |
| IL4 (ng/L) | 7 (14) | 0.22 (0.12–0.65) | 2 (4) | 0.41 (0.40–0.42) | 3 (6) | 0.18 (0.13–0.39) | 0.211* | – |
| IL10 (ng/L) | 2 (4) | 0.71 (0.37–1.04) | 2 (4) | 5.35 (4.56–6.15) | 5 (10) | 0.62 (0.53–1.11) | 0.433* | – |
| IL13 (ng/L) | 17 (33) | 0.21 (0.12–0.40) | 13 (25) | 0.26 (0.15–0.40) | 17 (33) | 0.31 (0.12–0.62) | 0.613 | 0.651 |
| IL17 (ng/L) | 5 (10) | 3.61 (1.01–4.88) | 0 | – | 0 | – | 0.011* | – |
| *Chemokines* | | | | | | | | | | |
| IL8 (ng/L) | 51 (100) | 15.35 (6.87–47.80) | 51 (100) | 11.24 (6.56–29.78) | 51 (100) | 14.11 (8.52–23.99) | 1.000 | 0.111 |
| MCP1 (ng/L) | 45 (88) | 249.82 (92.32–542.20)[a] | 30 (59) | 84.68 (27.16–179.62)[b] | 28 (55) | 33.51 (11.85–51.99)[b] | <0.001 | <0.001 |
| MIP1β (ng/L) | 49 (96) | 13.46 (4.78–39.98)[a] | 41 (80) | 5.75 (2.52–15.67)[b] | 36 (71) | 3.73 (2.03–7.79)[b] | 0.003 | <0.001 |
| GROα (µg/L) | 51 (100) | 1.69 (0.54–4.01)[a] | 51 (100) | 0.64 (0.23–1.99)[b] | 51 (100) | 0.57 (0.25–1.67)[b] | 1.000 | 0.004 |
| *Hematopoyetic factors* | | | | | | | | | | |
| IL5 (ng/L) | 24 (47) | 1.47 (0.81–2.02) | 24 (47) | 1.28 (0.87–1.89) | 25 (49) | 0.87 (0.65–1.66) | 0.975 | 0.363 |
| IL7 (ng/L) | 24 (47) | 101.55 (59.67–143.33) | 19 (37) | 42.92 (12.96–121.21) | 19 (37) | 82.87 (34.40–126.54) | 0.507 | 0.607 |
| GCSF (ng/L) | 15 (29) | 1.51 (1.26–3.04) | 16 (31) | 1.00 (0.54–3.69) | 18 (35) | 3.04 (1.50–3.90) | 0.811 | 0.505 |
| GMCSF (ng/L) | 0 | – | 0 | – | 0 | – | – | – |
| EGF (µg/L) | 51 (100) | 5.51 (4.37–6.78)[a] | 51 (100) | 4.84 (3.87–5.65)[b] | 51 (100) | 4.55 (3.47–5.42)[b] | 1.000 | <0.001 |
| TGFβ₂ (µg/L) | 51 (100) | 2.23 (1.33–3.30)[a] | 51 (100) | 1.30 (0.73–2.11)[b] | 51 (100) | 1.71 (0.87–2.63)[ab] | 1.000 | 0.016 |

**Abbreviations:** EGF, epidermal growth factor; GCSF, granulocyte colony-stimulating factor; GMCSF, granulocyte–macrophage colony-stimulating factor; Groα, growth-related oncogene-α; IFNγ, interferon-γ; Ig, immunoglobulin; IL, interleukin; MCP1, macrophage–monocyte chemoattractant protein-1; MIP1β, macrophage inflammatory protein-1β; TGFβ₂, transforming growth factor-β2; TNFα, tumor necrosis factor-α.

[a] n (%): number (percentage) of samples in which the immunological compound was detected.

[b] Concentrations are expressed as median and interquartile range (IQR).

[c] Chi-squared tests (or Fisher-tests, marked with an asterisk [*]) were used to evaluate differences in expression frequencies of the analyzed parameters across time.

[d] Friedman tests were used to evaluate differences in concentration along time. Different caption letters in a row mean statistical differences in the concentration of the immune factor between postnatal weeks when the *post hoc* pairwise comparison Nemenyi test was done.

## Concentrations and changes over time of immune factors

All the assayed immune factors could be detected among the milk samples provided by the 51 participants, with the only exception of GMCSF (Table 2). In general, the values obtained for all the immune factors showed a high inter-individual variability in detection frequencies (Table 2). Globally, IgA, IgGt, IgM, EGF, TGFβ2, IL8 and Groα displayed the highest detection frequencies (100% of the samples) at week 2, followed by IL1β, TNFα, MCP1 and MIP1β (75–97% of the samples). In contrast, IL2, IL10, IL12, IL17 and IFNγ were found in ≤ 10% of the samples.

When the detection frequencies of all immune factors at week 2 were compared with those obtained at weeks 6 and 12, statistical differences were observed for IL1β (p = 0.046), IL6 (p = 0.002), IL17 (p = 0.011), MCP1 (p < 0.001), and MIP1β (p = 0.003) (Table 2). The detection frequencies of these immune factors decreased from week 2 to week 6, and the reduction was particularly noticeable for MCP1 and IL6 (from 88% to 59% and from 55% to 31%, respectively). A further decrease in the detection frequency of MCP1 (to 55%), IL6 (to 22%), and MIP1β (71%) was observed at week 12. IL17 was not detected in any sample taken at week 6 or 12 (Table 2).

The concentrations of all detected immune factors also showed a high degree of variability between women, as reflected by the large IQR values for some of the analyzed parameters (Table 2). The most abundant immune factor detected at week 2 was IgA (median [IQR] = 1.68 [1.08–2.09] g/L), followed by IgM and IgGt (about 25 and 30 times less abundant, respectively). The second largest group of immune factors included hematopoietic factors EGF and TGFβ2 and the chemokine Groα with median concentrations of 5.51, 2.23 and 1.69 μg/L, respectively. The median levels of other immune factors ranged from 249.82 ng/L (for MCP1) to 0.21 ng/L (for IL13). Despite such inter-individual variability, statistically significant differences between postnatal weeks were found for the concentrations of IgA (p < 0.001), IgM (p < 0.001), IgGt (p < 0.001), IL6 (p = 0.002), MCP1 (p < 0.001), MIP1β (p < 0.001), GROα (p = 0.004), EGF (p < 0.001), and TGFβ2 (p = 0.016) (Table 2). The concentrations of these immune factors decreased over time, and the decline was particularly pronounced between samples taken at weeks 2 and 6. The only exception was TGFβ2: the median value decreased from week 2 (2.23 μg/L) to week 6 (1.30 μg/L) and increased from week 6 to week 12 (1.71 μg/L).

## Association between stress, anxiety and depressive symptoms and immune factor concentrations

Spearman's correlation coefficients between stress, anxiety, and depressive symptoms, and immune factors (only those that were detected in at least half of the samples) concentrations in milk at week 6 were very low, indicating very weak monotonic correlations between these variables (Table 3). Out of total 54 correlations analyzed, only three reached statistical significance: stress and IL8 ($r_s$ = 0.29, p = 0.039), depressive symptoms and IL7 ($r_s$ = 0.29, p = 0.040), and stress and TGFβ2 ($r_s$ = -0.34, p = 0.014) (Table 3). None of the psychosocial distress variables explained the concentrations of any of the immune factors according to the linear mixed effects models (results not shown).

## Association between stress, anxiety and depressive symptoms and cortisol concentrations

Linear mixed effects model allowed exploring the predictive values of stress, anxiety, and depressive symptoms, as well as postnatal week, sample collection time, and time elapsed between waking up and milk sampling on cortisol concentrations (Table 4). Cortisol concentrations in milk were higher with increasing stress (F = 5.28; p = 0.023) and with increasing anxiety (F = 6.51; p = 0.012), but they were not positively predicted by depressive symptoms. As described previously, cortisol concentrations in milk varied along postnatal weeks (F = 3.23; p = 0.043) and were strongly influenced by the sample collection time (F = 21.37; p = 0.000), but they were not related to the time elapsed between waking up and milk sampling (Table 4).

## Differences in concentrations of immune factors between mothers with low and high maternal psychosocial distress

In line with a previous study by Zijlmans et al. [36], we chose to compare women with highest and lowest psychosocial distress (i.e. compare the extremes within the study population). To

**Table 3. Correlation between immune factors concentrations and the postnatal psychosocial distress variables at week 6.**

| | Stress | | | Anxiety | | | Depressive symptoms | | |
|---|---|---|---|---|---|---|---|---|---|
| | $n^a$ | $r_s{}^b$ | p-value | n | $r_s$ | p-value | n | $r_s$ | p-value |
| *Immunoglobulins* | | | | | | | | | |
| IgA | 51 | 0.14 | 0.332 | 51 | 0.07 | 0.636 | 51 | -0.04 | 0.760 |
| IgGt | 51 | 0.19 | 0.187 | 51 | 0.07 | 0.610 | 51 | 0.09 | 0.533 |
| IgM | 51 | 0.25 | 0.082 | 51 | 0.11 | 0.429 | 51 | 0.19 | 0.177 |
| *Innate immunity* | | | | | | | | | |
| IL1β | 35 | 0.03 | 0.844 | 35 | 0.26 | 0.138 | 35 | 0.28 | 0.103 |
| TNFα | 34 | 0.05 | 0.777 | 34 | 0.23 | 0.197 | 34 | 0.05 | 0.764 |
| *Chemokines* | | | | | | | | | |
| IL8 | 51 | 0.29 | 0.039 | 51 | 0.14 | 0.343 | 51 | 0.03 | 0.820 |
| MCP1 | 30 | 0.07 | 0.718 | 30 | 0.04 | 0.818 | 30 | -0.09 | 0.640 |
| MIP1β | 41 | -0.15 | 0.351 | 41 | 0.06 | 0.704 | 41 | -0.28 | 0.081 |
| GROα | 51 | 0.03 | 0.837 | 51 | -0.17 | 0.221 | 51 | -0.15 | 0.286 |
| *Hematopoyetic factors* | | | | | | | | | |
| IL5 | 24 | -0.07 | 0.743 | 24 | -0.04 | 0.838 | 24 | -0.08 | 0.714 |
| IL7 | 51 | 0.23 | 0.109 | 51 | 0.10 | 0.498 | 51 | 0.29 | 0.040 |
| EGF | 51 | -0.01 | 0.939 | 51 | -0.13 | 0.368 | 51 | -0.04 | 0.794 |
| TGFβ₂ | 51 | -0.34 | 0.014 | 51 | 0.01 | 0.938 | 51 | -0.11 | 0.444 |

**Abbreviations:** EGF, epidermal growth factor; Groα, growth-related oncogene-α; Ig, immunoglobulin; IL, interleukin; MCP1, macrophage–monocyte chemoattractant protein-1; MIP1β, macrophage inflammatory protein-1β; TGFβ₂, transforming growth factor-β2; TNFα, tumor necrosis factor-α.

[a] n: sample size.

[b] $r_s$: Spearman's correlation coefficient.

this end, a median score for each of the three psychological variables was computed. The group of mothers who scored below the median on all three variables was categorized as 'low maternal postnatal psychosocial distress group' (group LOW) (n = 13, 25%). The group of mothers scoring above the median on all three variables was categorized as the 'high maternal postnatal psychosocial distress group' (group HIGH) (n = 13, 25%). There were no significant differences between women with LOW and HIGH psychosocial distress for age, educational background, infant sex, and gestational age at delivery (Table 5).

There were no differences in the detection frequency of any immune factor when both groups LOW and HIGH were compared at each postnatal week (week 2, 6 and 12) (Table 6). Regarding immune factor concentrations, significant differences were only found for IL4 (week 2), EGF (week 2), and IL5 (all three sampling times) (p < 0.05). Of note, these immune factors were only detected in a small percentage of the samples in both groups.

**Table 4. Linear mixed effects model of cortisol concentration (log transformed) in human milk.**

| Explanatory term | Sum squares | DF | Mean squares | F | p-value |
|---|---|---|---|---|---|
| Postnatal stress | 0.502 | 1 | 0.502 | 5.28 | 0.023 |
| Postnatal anxiety | 0.681 | 1 | 0.618 | 6.51 | 0.012 |
| Postnatal depressive symptoms | 0.059 | 1 | 0.059 | 0.62 | 0.434 |
| Postnatal week | 0.614 | 2 | 0.307 | 3.23 | 0.043 |
| Sample collection hour | 2.030 | 1 | 2.030 | 21.37 | 0.000 |
| Time elapsed between waking up and milk sampling | 0.0004 | 1 | 0.0004 | 0.00 | 0.947 |

**Table 5. Maternal and infant characteristics of participants with lowest (group LOW) and highest (group HIGH) levels of maternal postnatal psychosocial distress.**

| | Group LOW (n = 13) | Group HIGH (n = 13) | p-value |
|---|---|---|---|
| *Maternal characteristics* | | | |
| Age (years) | | | 0.159[d] |
| Mean (SD) | 30.50 (3.37) | 32.69 (4.09) | |
| Range (min, max) | 25–36 | 28–40 | |
| Educational background | | | |
| Middle job training, n (%) | 3 (23) | 1 (8) | 0.621[e] |
| Special college, n (%) | 3 (23) | 2 (15) | |
| University, n (%) | 7 (54) | 9 (69) | |
| Other, n (%) | 0 (-) | 1 (8) | |
| Psychosocial distress | | | |
| Postnatal stress[a] [median (IQR)] | 1.86 (1.60–2.00) | 2.50 (2.40–2.67) | < 0.001[f] |
| Postnatal general anxiety[b] [median (IQR)] | 24 (22–25) | 38.0 (32.0–43.0) | < 0.001 |
| Postnatal depressive symptoms[c] [median (IQR)] | 3.0 (0.0–6.0) | 10.0 (8.0–11.0) | < 0.001 |
| *Infant characteristics* | | | |
| Infant gender | | | |
| Boy, n (%) | 7 (54) | 6 (46) | 0.699[g] |
| Girl, n (%) | 6 (46) | 7 (54) | |
| Gestational age at birth (weeks) | | | |
| Mean (SD) | 39.95 (1.38) | 39.93 (1.28) | |
| Range (min, max) | 36.71–41.71 | 37.00–41.71 | 0.966[d] |

[a] Alledaagse Problemen Lijst (Everyday Problem Checklist; EPL; scale: 0–6) [29].

[b] State Trait Anxiety Inventory (STAI-S; scale: 20–80) [30].

[c] Edinburgh Postnatal Depression Scale (EPDS; scale: 0–30) [31].

[d] One-way ANOVA test was used to determine differences in gestational age at birth between groups LOW and HIGH.

[e] Fisher exact probability test was used to evaluate differences in mother's educational background between groups LOW and HIGH.

Chi-squared tests (or Fisher exact tests) were used to evaluate differences in the detection of the immune factors between groups LOW and HIGH at each sampling time and within each group across time. There were no differences in the detection frequency of any immune factor when both groups were compared at each sampling time. Statistical differences in the detection frequency of an immune factor within a group (LOW or HIGH) across time are indicated with a hash (♯ p< 0.05).

Mann-Whitney *U* tests were used to evaluate differences in the concentration of immune factors between groups LOW and HIGH at each sampling time. Statistical differences of the concentrations between groups LOW and HIGH are indicated with an asterisk (*p < 0.05).

Similar to the total group of women, the frequencies of detection of the immune factors decreased in both groups over time (Table 6). However, the change reached statistical significance only for IL6 in group LOW: IL6 was more frequently found among samples at week 2 (62%) than at weeks 6 and 12 (23%) (p = 0.044).

The concentrations of IgM, MCP1, and MIP1β decreased significantly in both groups over time (S1 Table), which was similar in comparison with the total group of participants (Table 2). There was a significant decrease in IgA (p = 0.002), IL8 (p = 0.012), and EGF (p = 0.023) concentrations in samples of mothers in group LOW, which was not observed in

**Table 6. Immune factors in milk samples provided by women with lowest (group LOW) and highest (group HIGH) maternal postnatal psychosocial distress.**

| | Week 2 | | | | Week 6 | | | | Week 12 | | | |
| | Group LOW (n = 13) | | Group HIGH (n = 13) | | Group LOW (n = 13) | | Group HIGH (n = 13) | | Group LOW (n = 13) | | Group HIGH (n = 13) | |
| | n[a] | Median (IQR) | n | Median (IQR) | n | Median (IQR) | n | Median (IQR) | n | Median (IQR) | N | Median (IQR) |
|---|---|---|---|---|---|---|---|---|---|---|---|---|
| *Immunoglobulins* | | | | | | | | | | | | |
| IgA, mg/L (×10³) | 13 | 1.8 (1.3–2.5) | 13 | 1.9 (1.1–2.1) | 13 | 1.5 (1.1–1.8) | 13 | 1.5 (1.0–2.0) | 13 | 1.2 (1.1–1.8) | 13 | 1.6 (1.1–1.7) |
| IgGt (mg/L) | 13 | 47.7 (34.6–85.9) | 13 | 52.1 (44.0–67.2) | 13 | 43.0 (30.4–58.4) | 13 | 36.4 (32.6–60.7) | 13 | 48.2 (37.3–51.7) | 13 | 42.0 (30.7–52.7) |
| IgM (mg/L) | 13 | 60.6 (54.7–93.1) | 13 | 78.7 (58.7–100.9) | 13 | 29.1 (18.4–55.3) | 13 | 45.3 (39.8–79.1) | 13 | 33.1 (25.4–39.9) | 13 | 42.9 (23.5–74.4) |
| *Innate immunity* | | | | | | | | | | | | |
| IL1β (ng/L) | 12 | 0.7 (0.2–2.6) | 12 | 0.5 (0.3–2.1) | 7 | 0.6 (0.3–0.79) | 10 | 0.7 (0.6–1.0) | 9 | 0.2 (0.1–0.3) | 10 | 0.4 (0.2–2.4) |
| IL6 (ng/L) | 8[♯] | 10.0 (6.8–16.3) | 7 | 7.8 (4.0–78.6) | 3[♯] | 4.6 (2.6–22.1) | 3 | 4.5 (3.6–7.5) | 3[♯] | 1.5 (1.2–2.1) | 4 | 2.7 (1.4–5.3) |
| IL12(p70) (ng/L) | 0 | - | 1 | 4.4 | 0 | - | 0 | - | 1 | 0.5 | 1 | 0.4 |
| IFNγ (ng/L) | 0 | - | 2 | 14.9 (11.2–18.5) | 0 | - | 0 | - | 0 | - | 1 | 2.1 |
| TNFα (ng/L) | 11 | 3.1 (1.4–6.1) | 10 | 2.0 (1.0–5.3) | 8 | 2.0 (0.9–3.0) | 10 | 3.0 (1.6–4.0) | 8 | 2.9 (0.7–5.0) | 9 | 2.8 (1.4–3.3) |
| *Acquired immunity* | | | | | | | | | | | | |
| IL2 (ng/L) | 0 | - | 1 | 0.9 | 0 | - | 0 | - | 0 | - | 2 | 3.4 (2.7–4.1) |
| IL4 (ng/L) | 3 | **0.50 (0.0–0.1)*** | 2 | **1.6 (1.2–1.9)*** | 0 | - | 0 | - | 1 | 0.2 | 1 | 0.1 |
| IL10 (ng/L) | 0 | - | 1 | 0.03 | 0 | - | 0 | - | 3 | 0.6 (0.3–0.9) | 1 | 0.5 |
| IL13 (ng/L) | 5 | 0.3 (0.3–0.7) | 6 | 0.2 (0.1–0.4) | 3 | 0.5 (0.3–0.7) | 6 | 0.26 (0.22–0.30) | 5 | 0.6 (0.2–0.8) | 3 | 0.6 (0.3–0.9) |
| IL17 (ng/L) | 2 | 2.6 (1.5–3.8) | 2 | 11.8 (7.7–16.0) | 0 | - | 0 | - | 0 | - | 0 | - |
| *Chemokines* | | | | | | | | | | | | |
| IL8 (ng/L) | 13 | 25.8 (11.0–83.0) | 13 | 13.3 (5.8–53.8) | 13 | 6.9 (4.5–27.6) | 13 | 8.7 (6.2–48.1) | 13 | 10.2 (9.3–17.1) | 13 | 14.5 (7.4–34.7) |
| MCP1 (ng/L) | 11 | 267.6 (125.9–700.8) | 10 | 372.2 (121.3–542.2) | 7 | 118.1 (90.2–218.6) | 5 | 110.1 (40.6–123.5) | 6 | 44.5 (35.6–53.2) | 8 | 43.1 (19.5–60.7) |
| MIP1β (ng/L) | 13 | 34.2 (7.7–68.1) | 13 | 14.7 (2.6–40.0) | 9 | 10.4 (4.5–15.7) | 10 | 5.3 (1.6–14.7) | 10 | 3.6 (1.8–4.0) | 9 | 2.6 (1.2–14.6) |
| GROα (µg/L) | 13 | 3.8 (1.1–4.1) | 13 | 4.1 (0.4–5.9) | 13 | 0.73 (0.2–6.4) | 13 | 0.3 (0.2–1.0) | 13 | 0.8 (0.5–2.1) | 13 | 0.4 (0.2–0.9) |
| *Hematopoyetic factors* | | | | | | | | | | | | |
| IL5 (ng/L) | 4 | **1.7 (1.1–2.6)*** | 6 | **1.6 (0.5–1.8)*** | 4 | **3.0 (2.4–3.4)*** | 6 | **1.3 (1.0–1.6)*** | 5 | **0.9 (0.8–0.9)*** | 5 | **1.5 (1.3–1.9)*** |
| IL7 (ng/L) | 5 | 134.0 (79.0–134.6) | 9 | 132.0 (90.7–145.2) | 5 | 119.2 (52.7–151.8) | 3 | 15.2 (13.5–57.7) | 4 | 137.5 (102.5–167.4) | 6 | 85.0 (59.1–162.1) |
| GCSF (ng/L) | 4 | 1.5 (1.4–3.1) | 5 | 2.3 (1.3–24.4) | 4 | 1.05 (0.7–3.9) | 3 | 0.1 (0.1–0.4) | 4 | 2.0 (1.6–2.6) | 6 | 3.0 (1.3–3.3) |
| GMCSF (ng/L) | 0 | - | 0 | - | 0 | - | 0 | - | 0 | - | 0 | - |
| EGF (µg/L) | 13 | **4.5 (3.6–5.2)*** | 15 | **5.8 (5.5–6.9)*** | 13 | 4.0 (3.6–4.9) | 13 | 5.3 (4.1–6.2) | 13 | 3.8 (2.7–5.4) | 13 | 4.6 (4.0–5.5) |
| TGFβ₂ (µg/L) | 13 | 2.12 (1.6–2.7) | 13 | 2.4 (1.3–3.7) | 13 | 1.2 (0.8–2.6) | 13 | 1.1 (0.7–1.5) | 13 | 2.2 (1.0–2.6) | 13 | 2.2 (1.7–2.7) |

**Abbreviations:** EGF, epidermal growth factor; GCSF, granulocyte colony-stimulating factor; GMCSF, granulocyte–macrophage colony-stimulating factor; Groα, growth-related oncogene-α; IFNγ, interferon-γ; Ig, immunoglobulin; IL, interleukin; MCP1, macrophage–monocyte chemoattractant protein-1; MIP1β, macrophage inflammatory protein-1β; TGFβ₂, transforming growth factor-β2; TNFα, tumor necrosis factor-α.

[a] n: number of samples in which the parameter was detected (relative frequency of detection).

women in group HIGH. In contrast, Groα only showed a significant decrease over time in group HIGH ($p < 0.001$) but not in group LOW (p = 0.926).

## Differences in cortisol concentrations between mothers with low and high maternal psychosocial distress

At week 2, the mean concentrations of cortisol in milk samples of group LOW were approximately less than half of those found in group HIGH [mean (95% CI) concentration of 6.62

**Table 7. Change in the cortisol concentration in milk samples, in the collection time and in the time elapsed between waking up and milk sampling in women with lowest (group LOW) and highest (group HIGH) maternal postnatal psychosocial distress.**

|  | Group LOW (n = 13) | Group HIGH (n = 13) |  |
|---|---|---|---|
| Postnatal week | Mean (95% CI) | Mean (95% CI) | p-value[a] |
| **Cortisol (µg/dL)** |  |  |  |
| Week 2 | 6.62 (4.36–8.88) | 14.10 (6.34–21.80) | 0.046 |
| Week 6 | 13.00 (7.03–19.10) | 8.73 (5.08–12.40) | 0.194 |
| Week 12 | 12.20 (7.81–16.60) | 18.60 (7.73–29.40) | 0.249 |
| **Collection time (h)** |  |  |  |
| Week 2 | 7.93 (6.70–9.16) | 8.37 (7.10–9.65) | 0.592 |
| Week 6 | 7.95 (6.89–9.00) | 9.35 (7.34–11.40) | 0.174 |
| Week 12 | 6.93 (5.98–7.88) | 9.71 (7.37–12.10) | 0.024 |
| **Time elapsed between waking up and milk sampling (min)** |  |  |  |
| Week 2 | 30.00 (8.58–51.40) | 63.40 (17.10–110.00) | 0.178 |
| Week 6 | 45.40 (15.20–75.60) | 154.00 (27.10–281.00) | 0.055 |
| Week 12 | 24.70 (2.05–47.40) | 153.00 (27.40–278.00) | 0.046 |

[a] *t*-tests were used to evaluate differences between both groups at each postnatal week.

(4.36–8.88) µg/dL and 14.10 (6.34–21.80) µg/dL, respectively; p = 0.046] (Table 7). Women in both groups collected their milk samples around the same time at week 2, with only a small gap of 18 min difference in the mean collection time (mean time group LOW 07:56 and group HIGH 08:14 am). However, an increasing trend in the difference in the collection time was observed along time, reaching 2 h and 46 min at week 12 (mean collection time was 06:56 for women in group LOW and 09:42 for women in group HIGH; p = 0.024). Therefore, it is possible that the difference in cortisol concentration in milk between women in group LOW [mean (95% CI) concentration of 12.20 (7.81–16.60) µg/dL] and group HIGH [mean (95% CI) concentration of 18.60 (7.73–29.40) µg/dL] at week 12 would have been greater if samples had been collected at similar day times. Additionally, the time-lapse between waking up and the moment in which women actually collected the sample also tended to increase in women of the HIGH group. Women in group LOW took the samples 33.6 min (mean time for the group) after waking up during the study while women in group H collected milk samples 120 min (mean time for the group) after waking up throughout the study. At week 2, the time passed from waking up to sample collection was approximately twice among women in group HIGH when compared to women in group LOW; this time-lapse increased to about 2 h at week 12 [mean (95% CI) elapsed time of 24.70 (2.05–47.40) min in group LOW and of 153 (27.40–278.00) min in group HIGH; p = 0.046] (Table 7).

## Discussion

The results of this study showed that most immune factors were present in the milk samples at the different stages of lactation and that their concentrations decreased over time. In our study group, which included healthy and mostly highly educated women, there was no apparent relationship between maternal psychosocial distress and immune factor concentrations, while maternal psychosocial distress at three months postpartum was related to higher milk cortisol concentrations. It must be highlighted that the women recruited in this study did not have high levels of psychosocial distress, as shown in Table 1.

## Changes in human milk immune factor concentrations over the first three months postpartum

The immune factors IgA, IgG, IgM, EGF, TGFβ2, GROα and IL8, which were present in all milk samples and detectable in all stages of lactation, have been previously described as the "core" immune factors in human milk in healthy individuals [1]. Concentrations of most immune factors decreased over the first three months postpartum, especially between weeks 2 and 6. Previous studies have shown that the concentrations of most immune factors are notably higher in colostrum compared to transient and mature milk [10,11]. Despite the observed decrease over time, the concentrations of all "core" immune factors remained within the previously reported ranges for healthy women [1,40]. The decrease in the concentrations of immune factors over time follows the diminishing needs of the developing infant for immune factors, due to the increased immune memory of the infant's own immune system (e.g., increased B cell repertoire development to produce IgA, IgM and IgG) [41]. This gradual decline parallels the decrease of neonatal Fc receptors (FcRn) located on enterocytes' surface (FcRn are receptors responsible for the uptake of Ig from gut lumen to infant blood), the infant's decreasing intestinal permeability to transfer maternal immune factors, as well as the increased acidity of the infant's stomach [41,42]. The idea that the composition of milk adjusts to the infant's needs is further supported by the parallel increase of energy and nutrients (e.g., lipids) in milk and the infant's increasing need for these components during the first weeks postpartum [43]. Our findings support the existing theory that maternal milk immune factors are especially important during the early postnatal period, compared to the late postpartum period, for optimal priming of the neonatal immune system [41,42].

## Human milk immune factor concentrations were not related to psychosocial distress

No significant association was found between psychosocial distress symptoms and the concentrations of immune factors in milk in this study. This finding is in contrast with previous studies that described positive correlations between immune factor concentrations in milk and depressive symptoms and/or anxiety [10,15,16]. Variations in the experimental designs, including sampling at different stages of lactation, may in part explain these conflicting results. For example, Thibeau et al. [10] found a negative association between maternal stress and EGF and MIP-1α in milk collected 3 days post-delivery, while we sampled later in the post-partum period. In addition, differences in maternal distress and characteristics between studies may also potentially explain discrepancies in results. Women in our sample scored lower on psychosocial distress compared to the other studies. Additionally, women included in this study were located in the Netherlands, reported no diseases and were highly educated, while mothers included in previous studies were located in the United States and Japan, reported sickness or allergic diseases, and included many mothers with lower socio-economic status [10,15,16]. Population characteristics, including geographical location, physical health, maternal lifestyle, and nutritional status, have been associated with changes in concentrations of milk immune factors [1,44]. For future research, it would be interesting to study a comparable population of Dutch women with higher levels of distress to enable greater understanding of the possible relation between maternal distress and milk immune factor concentrations.

## Human milk cortisol was related to maternal distress

In the present study, psychosocial distress was positively associated with increased milk cortisol concentrations. When investigating the relations between individual psychosocial distress

measures and milk cortisol concentrations, it appeared that while daily hassles and general anxiety were predictive of increased cortisol concentrations, depressive symptoms were not. Other studies that measured cortisol concentrations from the first days up till 12 weeks postpartum found a positive association between postpartum blues and higher levels of salivary and serum cortisol concentrations [25,27]. More specifically, Taylor et al. [27] reported a positive association for postpartum blues, but not for higher depressive symptoms, and serum cortisol concentrations. However, in contrast to our findings, two other studies reported that women with depressive symptoms, compared to women without symptoms, had significantly higher salivary and serum cortisol 14 days and 7 weeks post-partum [45,46]. Yet, several other studies indicated non-significant or negative associations between postpartum blues, depressive symptoms or anxiety and serum, salivary, or milk cortisol concentrations [3,47,48,49,50,51,52,53,54,55]. Finally, a recent study found no relation between a construct of maternal distress (i.e. depressive symptoms and anxiety) and milk cortisol concentrations at 3 months postpartum [3]. The large variability in experimental designs, methods, and outcomes of these studies hampers comparison of findings, making it difficult to draw definitive conclusions on possible associations between stress, anxiety, depressive symptoms, and milk cortisol concentrations. In order to compare studies and draw definitive conclusions, it would be worthwhile to initiate an international consensus expert meeting on the principles of milk research among researchers and clinicians in this field in order to create guidelines on milk collection methods, analyses techniques, and measurement outcomes that should be applied in future research.

Distinct cortisol concentrations in human milk, associated to different levels of psychosocial distress, could influence infant outcomes through breastfeeding. In line with the Lactational Programming hypothesis (i.e. the theory that biological components of milk can affect offspring neurobiological and metabolic development and in turn behavior and phenotype), these higher cortisol concentrations may have potential implications for infant health and behavior [56,57]. Milk cortisol is likely to be absorbed in the infant's gut [57], enter the systemic circulation, and cross the blood-brain barrier [58]. In the brain, cortisol can bind to glucocorticoid receptors in the limbic region, including the hippocampus and amygdala [59]. These brain areas are involved in, amongst others, behavior and emotion regulation [60]. Previous studies indeed suggest that exposure to higher levels of milk cortisol, together with other biologically active components and parenting behavior, may influence infant behavior [3,21]. For instance, milk cortisol has been related to infant experimentally induced emotional fear reactivity, although this behavior was only observed in girls. Next to the possible effects on infant behavior, milk cortisol might also influence maturation of the infant microbiome composition [61]. In this way, milk cortisol may indirectly affect the development of the infant's innate and adaptive immune systems, and contribute to the creation of a physical barrier to prevent pathogens from entering the body [62]. Moreover, milk cortisol is involved in the metabolism of carbohydrates, proteins and fats [63], and it may thus ensure sufficient energy uptake and weight gain in infants [2].

Better understanding of the biological significance of milk cortisol and its effects on both infant behavior and health is warranted, including estimations of the actual amount of milk cortisol that reaches the infant gut, blood circulation, and brain. Identifying the amount of cortisol that diffuses through the gut may help to further examine the potential effects of this hormone on infant health and behavior. Candidate methods to obtain this information would be to use *in vitro* evaluation methods to assess the passage of cortisol through the intestinal barrier [64], or *in vivo* rodent models with labelled cortisol to assess the amount of cortisol that reaches offspring blood circulation and brain [65,66].

Strengths of this study are its longitudinal design, the wide range of assayed milk immune factors, and the combined assessment of milk immune factors and cortisol concentrations. The main limitation of this study is that maternal psychosocial distress was measured only at one time point, while distress may change over time. Although some studies indicated that anxiety and depressive symptoms in the postpartum period are relatively constant [67,68], other studies reported that postpartum psychosocial distress in the late postpartum period is lower compared to the early postpartum period [69,70,71,72]. A next step in research would be to assess psychosocial distress at several time points throughout the postpartum period when investigating potential relations with milk immune factors and cortisol concentrations, because cortisol in human milk has a distinct circadian rhythm [73]. Although our study controlled for sample collection hour, women collected only one sample at each outcome assessment moment. Future studies should consider collecting multiple daily samples to provide more reliable information on the potential association between milk cortisol concentrations and psychosocial distress and to improve accuracy of results. These future studies should also standardize the daily time of collection. In this study, mothers were instructed to collect milk after waking up and before the first feed of infant. These instructions may have been too imprecise as reflected by the significant difference for time elapsed between waking up and sample collection between mothers with low and high psychosocial distress. Women with higher psychosocial distress reported later waking times, as well as longer periods between waking and sampling, over the weeks. As a result, and because milk cortisol concentrations are generally higher early in the morning, milk cortisol concentrations in women having high maternal psychosocial distress may have been underestimated in this study. Finally, women in our sample were predominantly highly educated and potential maternal confounders such as parity, atopy, BMI, health behavior (i.e. smoking, alcohol use, diet) were not accounted for, which may hamper the generalizability of the results. Future studies should consider including women clinically diagnosed with a stress or anxiety disorder, or with clinical depression, as well as increasing the sample size. When working with women with clinical diagnoses, extra attention should be given to instructions and reminders, in order to standardize sample collection times. Moreover, it would be relevant also to consider analyzing maternal intestinal microbiota since it may influence milk immune factors and bacterial composition [8,74]. Finally, research is needed to clarify whether varying concentrations of milk cortisol leave an imprint on infant health and development.

## Conclusions

In this study, we found a previously described "core" of milk immune factors that is naturally present in healthy mothers throughout the first three months postpartum. Previous reports have shown that immune factors in milk seem influenced by several factors, including maternal physical health, nutritional status, geographical location, and intestinal microbiota [1,42,74]. The current study added to this emerging literature by showing that immune factor concentrations in milk are not always associated to natural variations in maternal distress, at least in healthy populations such as the current one. This result should be confirmed using study populations with larger sample sizes, less uniformity of demographic characteristics, or higher (clinical) levels of anxiety, stress and/or depressive symptoms. Maternal psychosocial distress was positively related to higher cortisol concentrations in milk, which highlights the importance of further investigating this and other maternal factors in relation to human milk composition, as well as the potential effects of milk bioactive factors on infant health and behavior.

## Supporting information

**S1 Fig. Flowchart of the participants included in the study.**
(TIF)

**S2 Fig. Principal component analysis for the concentration of cortisol and immunological compounds in human milk samples (n = 152).** One sample was excluded because it determined the strong aggregation of the rest of the samples. Individual factor maps for milk samples, where each point represents a milk sample: **(A)** samples taken at week 2 postpartum are colored in green, at week 6 postpartum are colored in blue, and at week 12 postpartum are colored in red; **(B)** samples from women with low (0, red), medium (1, green) and high (2, blue) maternal postnatal psychosocial distress. **(C)** Correlation circle showing the active ($cos^2$>0.2) variables (arrows) contributing to the spatial distribution of the milk samples represented in the upper graph (A). Arrows pointing in the same direction are positively correlated, in the opposite direction are negatively correlated, and unrelated when they are orthogonal (90º).
(TIF)

**S3 Fig. Cortisol concentrations in human milk samples at week 2 (n = 50), week 6 (n = 51) and week 12 (n = 51) postpartum.** White circles represent outliers (>1.5× IQR). The bar on the top indicates significant differences between the cortisol concentration in milk samples taken at weeks 2 and 12 (Friedman's non parametric repeated measures comparison followed by *post-hoc* Nemenyi tests).
(TIF)

**S1 Table. Change in the concentrations of immune factors in milk samples of women categorized as having low (group LOW, n = 13) or high (group HIGH, n = 13) maternal postnatal psychosocial distress along time.**
(DOCX)

**S1 File. Dataset.**
(XLSX)

## Acknowledgments

The authors acknowledge and thank the contribution of all the participants who generously shared their time and samples for this study.

## Author Contributions

**Conceptualization:** Christine Hechler, Roseriet Beijers, Juan Miguel Rodríguez, Carolina de Weerth, Leonides Fernández.

**Data curation:** Marina Aparicio, Pamela D. Browne, Christine Hechler.

**Formal analysis:** Marina Aparicio, Pamela D. Browne, Leonides Fernández.

**Funding acquisition:** Roseriet Beijers, Juan Miguel Rodríguez, Carolina de Weerth, Leonides Fernández.

**Investigation:** Marina Aparicio, Pamela D. Browne, Christine Hechler.

**Methodology:** Marina Aparicio, Pamela D. Browne, Leonides Fernández.

**Project administration:** Juan Miguel Rodríguez, Carolina de Weerth.

**Resources:** Pamela D. Browne, Christine Hechler, Roseriet Beijers, Carolina de Weerth.

**Software:** Marina Aparicio.

**Supervision:** Juan Miguel Rodríguez, Carolina de Weerth, Leonides Fernández.

**Visualization:** Marina Aparicio, Pamela D. Browne, Leonides Fernández.

**Writing – original draft:** Marina Aparicio, Pamela D. Browne.

**Writing – review & editing:** Roseriet Beijers, Juan Miguel Rodríguez, Carolina de Weerth, Leonides Fernández.

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
