## [Decision Letter · Decision Letter 0]

16 Jan 2020

PONE-D-19-31037

Human milk cortisol and immune factors over the first three postnatal months: relations to maternal psychosocial distress

PLOS ONE

Dear Dr Fernández,

Thank you for submitting your manuscript to PLOS ONE. After careful consideration, we feel that it has merit but does not fully meet PLOS ONE’s publication criteria as it currently stands. Therefore, we invite you to submit a revised version of the manuscript that addresses the points raised during the review process.

Based on the comments of experts in the field, the manuscript presents several major weaknesses. These concern mainly the experimental design and methods -- especially statistical approach -- and undermine the relevance of the results obtained. A profound revision is needed to make the study more solid and reliable.

We would appreciate receiving your revised manuscript by Mar 01 2020 11:59PM. To enhance the reproducibility of your results, we recommend that if applicable you deposit your laboratory protocols in protocols.io, where a protocol can be assigned its own identifier (DOI) such that it can be cited independently in the future. For instructions see: http://journals.plos.org/plosone/s/submission-guidelines#loc-laboratory-protocols

We look forward to receiving your revised manuscript.

Kind regards,

Igor Branchi, Ph.D.

Academic Editor

PLOS ONE

Journal Requirements:

Reviewers' comments:

Reviewer's Responses to Questions

**Comments to the Author**

1. Is the manuscript technically sound, and do the data support the conclusions?

Reviewer #1: Partly

Reviewer #2: No

2. Has the statistical analysis been performed appropriately and rigorously? 

Reviewer #1: Yes

Reviewer #2: No

3. Have the authors made all data underlying the findings in their manuscript fully available?

Reviewer #1: Yes

Reviewer #2: Yes

4. Is the manuscript presented in an intelligible fashion and written in standard English?

Reviewer #1: Yes

Reviewer #2: Yes

5. Review Comments to the Author

Reviewer #1: The authors performed an interesting longitudinal investigation concerning how several active biological factors fluctuate throughout lactation (first three months) in human milk. The authors also attempted to correlate the fluctuations in these factors with perceived stress. The authors observed that while immune factors tended to decrease throughout lactation, cortisol concentrations increased over time and where correlated with self-reported psychological distress. The latter did not relate to maternal distress.

The manuscript is very well written and the authors are to be commended for being so transparent and honest with respect to the limitations of the study. I only have few minor issues that the authors may decide to take into account.

- The authors conclude that immune factors do not seem to correlate with maternal distress. This lack of effect may be due to a number of factors (many of which are also cogently discussed by the authors) that the experimental design did not allow taking into account. Whilst this does not devalue the study, I would simply suggest the authors to hedge this conclusion (see for example the abstract) and maybe clarify that the lack of association between distress and immune factors in milk may still be observed under other conditions (e.g. increased sample size, more uniform experimental population).

- Along the same considerations, I would suggest the authors to revise the second aim described in the introduction (lines 102-105). The part related to cortisol is fine, but maybe the part related to distress and immune functions shall be revised.

- Sometimes the authors did not specify what the meaning of numbers in parentheses. For example, on line 133, the authors report "... weeks [mean age of 14.75 (1.84)...]" what is 1.84? Is it standard error or standard deviation? Please specify.

- Still related to my principal concern, looking at data on distress (table 1) it seems to me that the subjects of the study were not particularly stressed. Could it be another reason why there does not seem to be a difference in immune response dependent on psychological distress? Please comment this aspect if deemed appropriate.

- I would suggest to put Fig S3 in the main text and not in the supplementary information.

- Line 463: the word postpartum reads odd. The font of the m is smaller that the other letters.

Reviewer #2: This manuscript assesses changes in breast milk immune factor and cortisol concentrations at postnatal weeks of 2, 6, and 12. Additionally, associations between maternal reports of psychological distress at 6weeks and the biological measures have been studied. The study population comprises of 55 healthy women. The topic is interesting and scarcely studied and the mechanisms via which maternal distress is transferred to the offspring also postnatally warrant attention. No child outcomes are included.

1. Considering the large number of biological measures that present with considerable normal variance, the study population is small. There is no power analysis, but there it is likely that the study has no sufficient power to answer the main questions.

2. Additional variance results by the fact that the sample collection time of the milk samples varies. This issue has been discussed and taken into account in the analyses but as the population is small, I wonder if this overcomes the acknowledged limitation of not having standardised sample collection times especially when it comes to cortisol concentrations.

3. The fact that psychological distress is actually a composite measure of conceptually different domains and there really are no subjects with actual high levels of symptoms is not solid enough. Investigating the influence of distress in a population of no distress is a limitation. The authors discuss this and describe the phenomenon as "normal variation in distress" but as the population is small, the milk samples come from three measurement points with varying sampling times and the distress symptoms have been only measured once at 6 weeks and then combined into this composite measure, I find it difficult to believe that the design is really strong in answering this particular question.

4. Distress symptoms ar measured cross-sectionally and inferences are made also backwards so that symptoms at 6 weeks are linked with milk cortisol at 2 weeks. Variation in symptom levels or persistence at a certain level in relation to variation in cortisol or immune factor levels would be more appropriate?

5. There is significant fluctuation in how cortisol levels seems to associate with high and low distress groups (where the numbers are 13/group, which appears small) depending by the time point of milk sampling. How do the authors interpret this (i.e. first the concentrations are higher, then lower and then again higher in high distress group vs the low distress group and all this happens within 10 weeks; Table 7). This, for example, underlines in my mind unreliability of the findings the unreliability being related to the underlying design and low number of subjects.

6. The authors should consider correction for multiple comparisons. In Table 7, for example, the result that is reported as one of the main outcomes is that cortisol levels in breast milk would be related to distress. However, the p value is close to 0.05, the groups are small, several tests have been conducted across the process and the direction of the associations changes between the three measurement points (#5) with only one measurement of the distress symptoms.

6. PLOS authors have the option to publish the peer review history of their article (what does this mean?). If published, this will include your full peer review and any attached files.

Reviewer #1: Yes: Simone Macrì

Reviewer #2: No

---

## [Author Response · Author response to Decision Letter 0]

6 Mar 2020

Reviewer #1: The authors performed an interesting longitudinal investigation concerning how several active biological factors fluctuate throughout lactation (first three months) in human milk. The authors also attempted to correlate the fluctuations in these factors with perceived stress. The authors observed that while immune factors tended to decrease throughout lactation, cortisol concentrations increased over time and where correlated with self-reported psychological distress. The latter did not relate to maternal distress.

The manuscript is very well written and the authors are to be commended for being so transparent and honest with respect to the limitations of the study. I only have few minor issues that the authors may decide to take into account.

1. The authors conclude that immune factors do not seem to correlate with maternal distress. This lack of effect may be due to a number of factors (many of which are also cogently discussed by the authors) that the experimental design did not allow taking into account. Whilst this does not devalue the study, I would simply suggest the authors to hedge this conclusion (see for example the abstract) and maybe clarify that the lack of association between distress and immune factors in milk may still be observed under other conditions (e.g. increased sample size, more uniform experimental population).

Answer: We agree with the reviewer and, therefore, the abstract and the conclusions sections have been modified as follows:

Abstract (conclusion subsection, lines 52-55 in the manuscript with tracked changes): “Immune factor concentrations in milk do not seem influenced by natural variations in maternal distress, at least in healthy populations. It is uncertain if this lack of association between immune factors in milk and maternal distress would also be observed in studies with larger populations, with less uniform demographic characteristics, or with women with higher (clinical) levels of anxiety, stress and/or depressive symptoms.”

Conclusions (lines 616-618): “The current study added to this emerging literature by showing that immune factor concentrations in milk do not seem influenced by natural variations in maternal distress, at least in healthy populations such as the current one. This result should be confirmed using study populations with larger sample sizes, less uniformity of demographic characteristics, or higher (clinical) levels of anxiety, stress and/or depressive symptoms.”

2. Along the same considerations, I would suggest the authors to revise the second aim described in the introduction (lines 102-105). The part related to cortisol is fine, but maybe the part related to distress and immune functions shall be revised.

Answer: The sentence has been modified as follows (lines108-109):

“The second aim was to identify whether maternal postnatal psychosocial distress (i.e. perceived stress due to daily hassles, general anxiety, and depressive symptoms) in a group of 51 healthy mothers was related to changes in the presence and concentration of immune factors and cortisol in human milk”. 

3. Sometimes the authors did not specify what the meaning of numbers in parentheses. For example, on line 133, the authors report "... weeks [mean age of 14.75 (1.84)...]" what is 1.84? Is it standard error or standard deviation? Please specify.

Answer: We apologize for the omission. It refers to the standard deviation (SD) and it has been indicated in the revised manuscript (lines137-142):

“Mothers collected milk after the infant reached the age of 2 weeks [mean (SD) age of 14.75 (1.84) days], 6 weeks [mean (SD) age of 43.58 (5.02) days], and 12 weeks [mean (SD) age of 85.35 (2.33) days]. Prior to collection, mothers washed their hands, breasts, and nipples, and cleaned the small collection cups with boiling water. Subsequently, prior to feeding the infant, they collected 20 mL of the first foremilk in the morning [mean (SD) time at 08:36 (02:48) am] by hand expression.”

4. Still related to my principal concern, looking at data on distress (table 1) it seems to me that the subjects of the study were not particularly stressed. Could it be another reason why there does not seem to be a difference in immune response dependent on psychological distress? Please comment this aspect if deemed appropriate.

Answer: We are aware of this limitation, and, in fact, we had indicated this concern in some parts of the original manuscript (for example, in the discussion section within the subsection "Human milk immune factor concentrations were not related to psychosocial distress”). However, we agree that this issue is very relevant and, as a consequence (and to further clarify this point), we have highlighted this more in the revised manuscript, as follows:

Abstract (conclusion) (lines 51-55): “Immune factor concentrations in milk do not seem influenced by natural variations in maternal distress, at least in healthy populations. It is uncertain if this lack of association between immune factors in milk and maternal distress would also be observed in studies with larger populations, with less uniform demographic characteristics, or with women with higher (clinical) levels of anxiety, stress and/or depressive symptoms.” 

Discussion (lines 470-475): “In our study group, which included healthy and mostly highly educated women, there was no apparent relationship between maternal psychosocial distress and immune factor concentrations, while maternal psychosocial distress at three months postpartum was related to higher milk cortisol concentrations. It must be highlighted that the women recruited in this study did not have high levels of psychosocial distress, as shown in Table 1.” 

Conclusions (lines 613-618): “The current study added to this emerging literature by showing that immune factor concentrations in milk do not seem influenced by natural variations in maternal distress, at least in healthy populations such as the current one. This result should be confirmed using study populations with larger sample sizes, less uniformity of demographic characteristics, or higher (clinical) levels of anxiety, stress and/or depressive symptoms.” 

5. I would suggest to put Fig S3 in the main text and not in the supplementary information.

Answer: We included that figure as supplementary material because of the fact that the cortisol concentration increases with the lactation time had been already reported (ref. 20. Hechler C, Beijers R, Riksen-Walraven JM, de Weerth C. Are cortisol concentrations in human breast milk associated with infant crying? Dev Psychobiol. 2018;60(6): 639–650. doi: 10.1002/dev.21761). We therefore think it is more adequate to present it as supplementary material but if the reviewer considers that it is a requisite for manuscript approval, then we would have no problem to include it in the revised manuscript. 

6. Line 463: the word postpartum reads odd. The font of the m is smaller that the other letters.

Answer: We have corrected the mistake (line 477).

Reviewer #2: This manuscript assesses changes in breast milk immune factor and cortisol concentrations at postnatal weeks of 2, 6, and 12. Additionally, associations between maternal reports of psychological distress at 6weeks and the biological measures have been studied. The study population comprises of 55 healthy women. The topic is interesting and scarcely studied and the mechanisms via which maternal distress is transferred to the offspring also postnatally warrant attention. No child outcomes are included.

Answer: We thank you for your comment on the innovative aspect of this study. Some child outcomes in this study group, such as the measurement of infant fussing and crying, are described in ref. 20 (Hechler C, Beijers R, Riksen-Walraven JM, de Weerth C. Are cortisol concentrations in human breast milk associated with infant crying? Dev Psychobiol. 2018;60(6): 639–650. doi: 10.1002/dev.21761). 

1. Considering the large number of biological measures that present with considerable normal variance, the study population is small. There is no power analysis, but there it is likely that the study has no sufficient power to answer the main questions.

Answer: We agree that it would have been desirable to have a larger group of participants, but our final experimental group comprises 51 women (out of an initial number of 88) who provided samples of milk during the postnatal period and filled out a questionnaire on psychosocial stress, anxiety and depressive symptoms. However, because we have 3 assessment moments for each of the 51 participants, we consider the main results of our work to be valid. These results may be seen as first evidence or preliminary results that will be very valuable to design future studies related to the influence of maternal psychosocial distress and, mainly, cortisol. 

One of the main results is the relationship between milk cortisol concentration and maternal psychosocial distress, that it is summarized in Table 4. This table also includes other factors that may have influence in the cortisol concentration (such as the postnatal week, the sample collection hour, and time elapsed between waking up and milk sampling). This relationship was explored using a linear mixed effects model, as indicated in the Statistical analysis section.

Additionally, and in accordance with the suggestion of the reviewer, we conducted a power analysis calculation for the main results presented in the manuscript. The power analysis of the linear mixed effects model of cortisol concentration in human milk given in Table 4 was performed using the G*Power version 3.1.9.2 (Faul, F; Universität Kiel, Germany). The squared multiple correlation R2 = 0.224 gives an effect size f2 = 0.289 (medium-large), resulting in a power of 0.9997 for a sample size of 153. Therefore, it can be inferred that the study has enough power. 

2. Additional variance results by the fact that the sample collection time of the milk samples varies. This issue has been discussed and taken into account in the analyses but as the population is small, I wonder if this overcomes the acknowledged limitation of not having standardised sample collection times especially when it comes to cortisol concentrations.

Answer: We are aware of the importance of the sampling time when determining cortisol concentrations in a biological sample, a fact that it is explicitly acknowledged in Figure 2. Since differences were not found when the distribution of the samples taken at the three sampling points (week 2, 6, and 12) was analyzed, we ruled out that the increase of cortisol concentration in milk over time was due to differences in the collection time. 

Participants were instructed to take the milk sample after getting up in the morning and before feeding the baby. However, inspection of the sampling times revealed that some women had taken samples at later times. When we compared mothers with lower and higher maternal psychosocial distress (always taking into account that all of them were healthy mothers), we realized that the group of women with the highest maternal psychosocial distress collected their samples increasingly late while the opposite was observed for the group of women with the lowest maternal psychosocial distress. This difference may possibly be related to their psychosocial distress, e.g. with women with more psychosocial complaints getting up later or organizing themselves less efficiently. As a result, and as it has been pointed out in the discussion, cortisol concentrations in milk from the group of women with the highest maternal psychosocial distress may have been underestimated. This may also be a useful observation for future studies in this field and that is why we recommend that “future studies should also standardize the daily time of collection” (line 592). 

3. The fact that psychological distress is actually a composite measure of conceptually different domains and there really are no subjects with actual high levels of symptoms is not solid enough. Investigating the influence of distress in a population of no distress is a limitation. The authors discuss this and describe the phenomenon as "normal variation in distress" but as the population is small, the milk samples come from three measurement points with varying sampling times and the distress symptoms have been only measured once at 6 weeks and then combined into this composite measure, I find it difficult to believe that the design is really strong in answering this particular question.

Answer: Yes, we agree that this is a complex condition, with a variety of levels, ranging from a mild to a very severe clinical state. This study reflects such complexity, including some difficulties found during the course of the study. Although the design is far from perfect, we sincerely think that it provides valuable information in the field: (a) by showing the lack of an association between a large set of immune markers and variations in maternal distress in a healthy population; (b) by showing an association between maternal distress and milk cortisol in samples from healthy women; and (c) by providing useful information about the evolution of immune parameters in human milk along time. In addition, and as stated by the reviewer, it contains useful information to improve the design of future studies and to avoid that other scientists may face similar problems. As usually reminded by ethical committees, the results of clinical studies are useful even when they fail to find clear correlations or when they show design inconsistencies that should be improved in future works.

4. Distress symptoms are measured cross-sectionally and inferences are made also backwards so that symptoms at 6 weeks are linked with milk cortisol at 2 weeks. Variation in symptom levels or persistence at a certain level in relation to variation in cortisol or immune factor levels would be more appropriate?

Answer: We agree with the reviewer that it would be preferable to have information about the evolution of the maternal psychosocial distress during the whole study period. Unfortunately, we only had access to information at that specific moment (1.5 months postpartum). However, we do not expect large intra-individual variations in psychosocial distress from 2 to 12 weeks postpartum. Intra-individual levels of psychosocial distress measures have been found to be stable over longer periods of time, from pregnancy to 6 weeks postpartum, and from 6 weeks postpartum till 2 years postpartum (Dipietro JA1, Costigan KA, Sipsma HL. Continuity in self-report measures of maternal anxiety, stress, and depressive symptoms from pregnancy through two years postpartum. J Psychosom Obstet Gynaecol. 2008 Jun;29(2):115-24). 

5. There is significant fluctuation in how cortisol levels seems to associate with high and low distress groups (where the numbers are 13/group, which appears small) depending by the time point of milk sampling. How do the authors interpret this (i.e. first the concentrations are higher, then lower and then again higher in high distress group vs the low distress group and all this happens within 10 weeks; Table 7). This, for example, underlines in my mind unreliability of the findings the unreliability being related to the underlying design and low number of subjects.

Answer: It is true that there is a wide variation in the cortisol levels, and that the two selected groups with low and high maternal psychosocial distress have a low number of individuals. Nevertheless, even with those drawbacks the information shown in Table 7 shows a tendency to different behavior (collection time and time elapsed between waking up and milk sampling) in both groups of women. In order to cope with the reviewer's observation, the whole paragraph has been rewritten in the revised manuscript (lines 432-453):

“At week 2, the mean concentrations of cortisol in milk samples of group LOW were approximately less than half of those found in group HIGH [mean (95% CI) concentration of 6.62 (4.36-8.88) μg/dL and 14.10 (6.34-21.80) μg/dL, respectively; p = 0.046] (Table 7). Women in both groups collected their milk samples around the same time at week 2, with only a small gap of 18 min difference in the mean collection time (mean time group LOW 07:56 and group HIGH 08:14 am). However, an increasing trend in the difference in the collection time was observed along time, reaching 2 h and 46 min at week 12 (mean collection time was 06:56 for women in group LOW and 09:42 for women in group HIGH; p = 0.024). Therefore, it is possible that the difference in cortisol concentration in milk between women in group LOW [mean (95% CI) concentration of 12.20 (7.81-16.60) μg/dL] and group HIGH [mean (95% CI) concentration of 18.60 (7.73-29.40) μg/dL] at week 12 would have been greater if samples had been collected at similar day times. Additionally, the time-lapse between waking up and the moment in which women actually collected the sample also tended to increase in women of the HIGH group. Women in group LOW took the samples 33.6 min (mean time for the group) after waking up during the study while women in group H collected milk samples 120 min (mean time for the group) after waking up throughout the study. At week 2, the time passed from waking up to sample collection was approximately twice among women in group HIGH when compared to women in group LOW; this time-lapse increased to about 2 h at week 12 [mean (95% CI) elapsed time of 24.70 (2.05-47.40) min in group LOW and of 153 (27.40-278.00) min in group HIGH; p = 0.046)] (Table 7).” 

6. The authors should consider correction for multiple comparisons. In Table 7, for example, the result that is reported as one of the main outcomes is that cortisol levels in breast milk would be related to distress. However, the p value is close to 0.05, the groups are small, several tests have been conducted across the process and the direction of the associations changes between the three measurement points (#5) with only one measurement of the distress symptoms.

Answer: We agree that some p-values shown in Table 7 are close to 0.05, which is an arbitrary set point for significance. We did not correct for multiple comparisons because data in Table 7 correspond to two independent groups and three different variables that had been examined at three different sampling times. According to the reviewer’s suggestion, we lowered the intensity of the association and indicated that the results just show a tendency, and instead of pointing out to the statistically significant differences, we indicated the raw values (mean and 95% confidence interval) to show their wide variation of values, and the p-value obtained in the contrast, as indicated in the previous question.

---

## [Decision Letter · Decision Letter 1]

8 Apr 2020

PONE-D-19-31037R1

Human milk cortisol and immune factors over the first three postnatal months: relations to maternal psychosocial distress

PLOS ONE

Dear Dr Fernández,

Thank you for submitting your manuscript to PLOS ONE. After careful consideration, we feel that it has merit but does not fully meet PLOS ONE’s publication criteria as it currently stands. Therefore, we invite you to submit a revised version of the manuscript that addresses the points raised during the review process.

The manuscript has been improved by the authors. However, there are still some minor points that need to be revised according to the Reviewers' suggestion. These concern mainly the Discussion section and the conclusions drawn.

We would appreciate receiving your revised manuscript by May 23 2020 11:59PM. To enhance the reproducibility of your results, we recommend that if applicable you deposit your laboratory protocols in protocols.io, where a protocol can be assigned its own identifier (DOI) such that it can be cited independently in the future. For instructions see: http://journals.plos.org/plosone/s/submission-guidelines#loc-laboratory-protocols

We look forward to receiving your revised manuscript.

Kind regards,

Igor Branchi, Ph.D.

Academic Editor

PLOS ONE

Reviewers' comments:

Reviewer's Responses to Questions

**Comments to the Author**

1. If the authors have adequately addressed your comments raised in a previous round of review and you feel that this manuscript is now acceptable for publication, you may indicate that here to bypass the “Comments to the Author” section, enter your conflict of interest statement in the “Confidential to Editor” section, and submit your "Accept" recommendation.

Reviewer #1: All comments have been addressed

Reviewer #2: (No Response)

2. Is the manuscript technically sound, and do the data support the conclusions?

Reviewer #1: (No Response)

Reviewer #2: Partly

3. Has the statistical analysis been performed appropriately and rigorously? 

Reviewer #1: (No Response)

Reviewer #2: Yes

4. Have the authors made all data underlying the findings in their manuscript fully available?

Reviewer #1: (No Response)

Reviewer #2: Yes

5. Is the manuscript presented in an intelligible fashion and written in standard English?

Reviewer #1: (No Response)

Reviewer #2: Yes

6. Review Comments to the Author

Reviewer #1: (No Response)

Reviewer #2: The authors have sufficiently acknowledged most of the comments.

I still have two suggestions to complete the revision:

1. Absence of evidence is not evidence of absence. Considering all the limitations related to the sample size and the actual design, I would like the conclusions to be phrased differentially. Instead of stating that "Immune factor concentrations in milk do not seem influenced by natural variations in maternal distress" I would like to see a comment statin that "We found no evidence on ... " or "Our data did not support ...." There are so many limitations and restrictions underlying this paper, I would like the conclusion to comprise more directly the many uncertainties related to the absence of the expected association.

2. The implications of my point #5 should also be discussed in the discussion section in addition to the revised results section. the fact that the important variation in sampling time was biased by the maternal condition of psychological symptoms should be discussed clearly.

7. PLOS authors have the option to publish the peer review history of their article (what does this mean?). If published, this will include your full peer review and any attached files.

Reviewer #1: Yes: Simone Macrì

Reviewer #2: Yes: Linnea Karlsson

---

## [Author Response · Author response to Decision Letter 1]

7 May 2020

We are glad the reviewers found our revised manuscript (almost) ready for publication.

Below we answer Reviewer 2’s remaining two points.

Reviewer #2: The authors have sufficiently acknowledged most of the comments.

I still have two suggestions to complete the revision:

1. Absence of evidence is not evidence of absence. Considering all the limitations related to the sample size and the actual design, I would like the conclusions to be phrased differentially. Instead of stating that "Immune factor concentrations in milk do not seem influenced by natural variations in maternal distress" I would like to see a comment statin that "We found no evidence on ... " or "Our data did not support ...." There are so many limitations and restrictions underlying this paper, I would like the conclusion to comprise more directly the many uncertainties related to the absence of the expected association.

We agree with the reviewer and replaced our concluding sentence in the Abstract by: “In the current study we found no evidence for an association between natural variations in maternal distress and immune factor concentrations in milk.” (lines 51-52)

We additionally adapted the sentence starting on line 607 of the Discussion as follows: “The current study added to this emerging literature by showing that immune factor concentrations are not always associated to natural variations in maternal distress, at least in healthy populations such as the current one.” This sentence replaces the sentence: “The current study added to this emerging literature by showing that immune factor concentrations in milk do not seem influenced by natural variations in maternal distress, at least in healthy populations such as the current one.”

2. The implications of my point #5 should also be discussed in the discussion section in addition to the revised results section. the fact that the important variation in sampling time was biased by the maternal condition of psychological symptoms should be discussed clearly.

We now adapted the sentence starting on line 590 to “Women with higher psychosocial distress reported later waking times, as well as longer periods between waking and sampling, over the weeks. As a result, and because milk cortisol concentrations are generally higher early in the morning, milk cortisol concentrations in women having high maternal psychosocial distress may have been underestimated in this study.”

We additionally added the following recommendation for future studies on line 597: “When working with women with clinical diagnoses, extra attention should be given to instructions and reminders, in order to standardize sample collection times.”

---

## [Editor Report · Decision Letter 2]

8 May 2020

Human milk cortisol and immune factors over the first three postnatal months: relations to maternal psychosocial distress

PONE-D-19-31037R2

Dear Dr. Fernández,

We are pleased to inform you that your manuscript has been judged scientifically suitable for publication and will be formally accepted for publication once it complies with all outstanding technical requirements.

With kind regards,

Igor Branchi, Ph.D.

Academic Editor

PLOS ONE
---

## [Editor Report · Acceptance letter]

13 May 2020

PONE-D-19-31037R2 

Human milk cortisol and immune factors over the first three postnatal months: relations to maternal psychosocial distress 

Dear Dr. Fernández:

I am pleased to inform you that your manuscript has been deemed suitable for publication in PLOS ONE. Congratulations! Your manuscript is now with our production department. 

With kind regards,

on behalf of

Dr. Igor Branchi 

Academic Editor

PLOS ONE